# Evolving LLM-Based Schemas for Mid-Vision Feedback

## Abstract

In this work, we present **ELF** (**E**volving **L**LM-Based Schemas for Mid-Vision **F**eedback), a framework that integrates schema evolution with Mid Vision Feedback (MVF) for visual learning. We leverage Large Language Models (LLMs) to automatically generate schemas: executable semantic programs operating over sets of context categories (e.g., "animate" or "inanimate"). We integrate schemas into visual processing via MVF, a method that utilizes top-down feedback connections to inform mid-level visual processing with high-level contextual knowledge. To optimize these schemas we utilize EvoPrompt, an evolutionary algorithm that refines schemas through iterative search, resulting in improvements in accuracy and contextual consistency. We demonstrate the effectiveness of ELF across multiple datasets and multiple architectures for the task of object classification.

## 1 Introduction

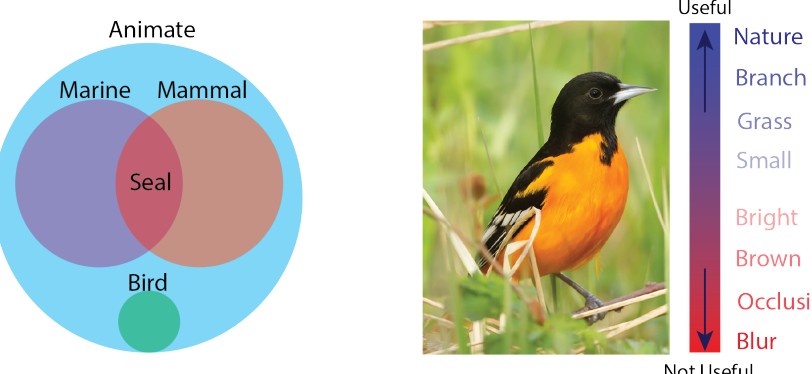

Figure 1: A) Contexts within a schema have various relations (e.g., containment, overlap, exclusivity) with other contexts defined within the schema. A schema might, for instance, include animate and inanimate contexts, with a subset of animate context relations shown here. B) Some contexts will be more useful to vision processing than others. This may be based in part on how reliably that context can be visually detected, and how that context interacts with the object categories.

Feedback plays a primary role in biological vision; in fact, the majority of neural connections in the visual cortex are top-down, rather than bottom-up, connections (Markov et al., 2014). These top-down connections are thought to convey information of higher level expectation, and neurons of the visual cortex use both higher level expectation as well as lower level visual information in producing their representations.

We employ in this work a mechanism - Mid-Vision Feedback (MVF) (Maynord et al., 2023) - which emulates the feedback systems in biological vision, where high-level context informs lower-level visual processing. Unlike traditional feed-forward architectures, where information flows only from low-level pixels to high-level abstract concepts, MVF introduces top-down feedback connections that integrate contextual knowledge into mid-level visual processing - see Figure 2. In the MVF framework, specific affine transformations (i.e., linear transformation and possible bias) are applied

to feature vectors at mid-level layers of a artificial neural network, based on the expected context (e.g., "animate" vs. "inanimate"). This contextual "hint" improves classification performance of the network.

Schemas are cognitive structures that help individuals organize and interpret information based on prior knowledge and experiences (Rumelhart, 2017; Axelrod, 1973). They influence how we perceive new stimuli by providing a framework for understanding and responding to the world around us. In biological systems, schemas help filter and organize sensory information, enabling faster and more accurate recognition by linking incoming stimuli to expected patterns (Bar, 2007).

While this process of applying high level contextual understanding to lower level processes is afforded significant resources in biological vision, contemporary vision architectures by-and-large do not involve similar dynamics. In artificial vision systems, leveraging schemas derived from language or contextual knowledge can allow the model to impose meaningful constraints on mid-level visual representations. These constraints could reduce noise, enhance feature relevance, and enable more precise predictions by narrowing the focus to contextually significant visual features. By aligning mid-level representations with higher-order schemas, vision models can improve generalization and efficiency, particularly in environments with numerous or overlapping categories. While schemas are of use across a variety of vision tasks, for the purposes of this work, we apply ELF to object classification.

Contexts in object detection can enhance a classifier's ability to distinguish different objects by providing additional relational information. These contexts are linked to the dataset's classes and the broader world, an example context being the distinction between animate and inanimate objects. See Figure 1 for an example context set and relations. Within the animate category, ontological structures like mammals, marine animals, and birds can be introduced, with relations that sometimes overlap (e.g., seals as marine mammals) but remain distinct from others (e.g., birds have no overlap with marine animals). These contexts exhibit different visual characters — e.g., mammals tend to have with skin and four limbs, while birds have feathers — that can bias object detection by guiding the classifier. The space of possible contexts which might be considered can be quite large.

We derive an initial unrefined context set from a Large Language Model (LLM), which generates a hierarchical ontology reflecting the structure of the visual world. The production of the schema is shown by the LLM to Schema arrow in Figure 3. By providing the LLM with an ontology of classes from a vision dataset and a prompt describing the learning task, the LLM maps each class to a set of class-derived contexts that aid in distinguishing between classes. This automated approach, instead of relying on human engineering, reduces labor overhead and avoids potential biases introduced by manual context creation.

From this initial context set, the LLM derives a set of initial schemas. See the schema within Figure 3 for a simplified illustration. ELF schemas essentially act as a decision-making engine that interprets visual data based on contexts and applies them to modulate the underlying feature representations in the vision network. The goal is to ensure that object predictions are not just based on raw visual features but are guided by meaningful, high-level context associations.

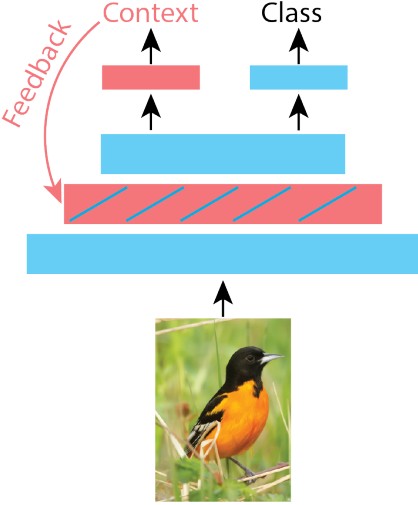

Figure 2: Illustration of feedback within MVF. This consists of a conventional architecture run in a feedforward fashion for object classification - this involves arrows shown in black. In addition to object classification, a context head is added to the output of the network, shown in red. This context prediction is then **fed back** to an intermediate layer of the network, and integrated with that layer's feature representations. The network is then rerun with this contextual biasing incorporated, enhancing object classification performance.

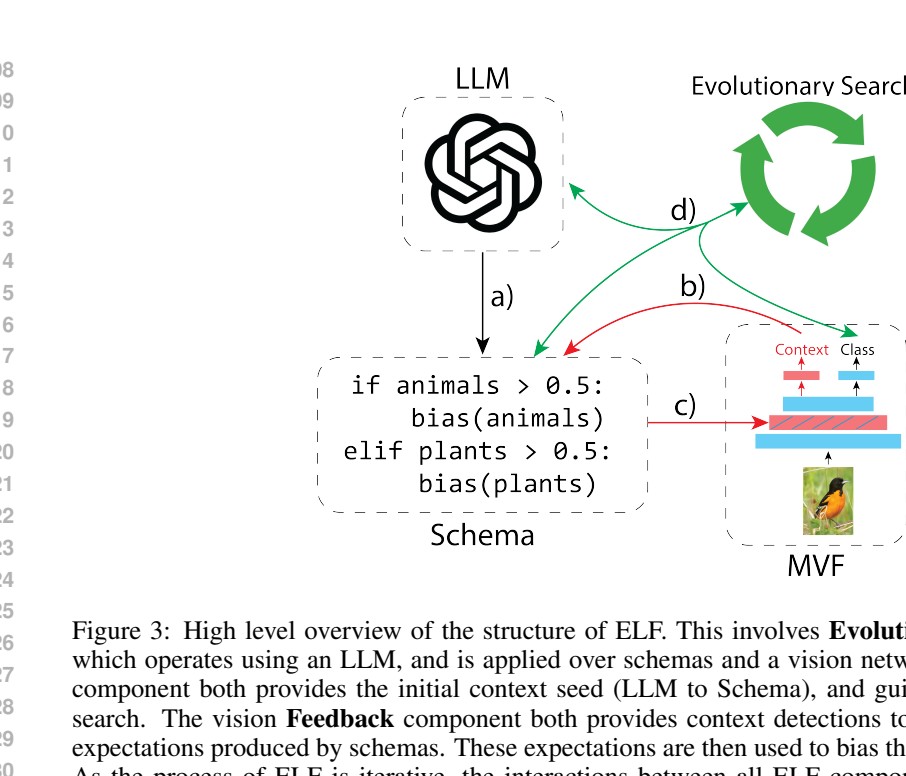

Figure 3: High level overview of the structure of ELF. This involves **Evolution**, shown in green, which operates using an LLM, and is applied over schemas and a vision network. The **Language** component both provides the initial context seed (LLM to Schema), and guides the evolutionary search. The vision **Feedback** component both provides context detections to schemas, and takes expectations produced by schemas. These expectations are then used to bias the perceptual process. As the process of ELF is iterative, the interactions between all ELF components involve cycles. Connection **a** seeds the schema set with an initial LLM derived seed set. Connection **b** feeds detected visual context into the schema. Connection **c** uses schema information in biasing the vision network. Connection **d** applies an evolutionary process over ELF components, employing an LLM, and updating schemas and the vision component.

As the exploratory space for schemas is large, we employ EvoPrompt (Chen et al., 2024). EvoPrompt is conventionally used in architecture search; in this work we use it in schema search. EvoPrompt facilitates the exploratory search by leveraging an evolutionary approach with an LLM crossover function to generate and refine candidate schemas. The relation between ELF components and evolutionary search is illustrated in green in Figure 3.

Candidate schemas produced by EvoPrompt capture a hierarchy of representation. This is integrated into visual processing through Mid-Vision Feedback (MVF), a feedback mechanism that maps the hierarchical language-derived representations onto visual features. This process, illustrated by the red arrows in Figure 3, allows the LLM to adjust visual processing by aligning representations with task-relevant cues. By incorporating language-driven contextual adjustments, ELF produces improvements to both accuracy and contextual consistency.

## 1.1 CONTRIBUTIONS

The primary contributions of this work are as follows:

- We introduce a method for the automatic production of **schemas**: executable semantic programs and ranking over contexts, generated via LLM and evolutionary search.

- We integrate vision and language through a novel, biologically inspired approach, using **Mid-Vision Feedback** to apply hierarchical structures from language to the vision system. This differentiates our work from flat-hierarchy models commonly used in vision research.

- We demonstrate the utility of ELF over multiple datasets and architectures. We will publicly release schemas which have been demonstrated to be transferable across architecture types, architecture sizes, and across related datasets. We will also release ELF, which allows for the production of schemas for arbitrary domains.

## 2 RELATED WORK

### 2.1 TOP-DOWN METHODS IN COMPUTER VISION

Previous works have explored the significance of feedback in biological sensory perception and its parallels in computer vision (Kveraga et al., 2007; Markov et al., 2014; Gilbert & Sigman, 2007; Kreiman & Serre, 2020; Gilbert & Li, 2013; Paneri & Gregoriou, 2017). Feedback models have also been shown to better organize mid-level visual features like animacy vs. inanimacy (Long et al., 2018) and texture representation (Jagadeesh & Gardner, 2022), aligning more closely with human perception (Harrington & Deza, 2021). In existing computer vision methods, feedback is typically implemented through recurrent connections (Caswell et al., 2016; Pinheiro & Collobert, 2014; Zamir et al., 2016), but alternatives like hierarchical rectified Gaussians (Hu & Ramanan, 2016) and graphical models (Yao et al., 2012) have been explored to enable top-down and bottom-up information flow, improving tasks such as keypoint localization and scene understanding. In this work we adapt the approach from Mid-Vision Feedback (Maynord et al., 2023), an emulation of the feedback systems in biological vision, where high-level context informs lower-level visual processing.

### 2.2 EVOLUTIONARY METHODS

Evolutionary algorithms have been widely used in various machine learning tasks (Chen et al., 2019; Lopes et al., 2022; Zhou et al., 2021; Zhang et al., 2021), particularly in neural architecture search (NAS) and optimization (Elsken et al., 2018; Ci et al., 2021; Zhu et al., 2019). EvoPrompting (Chen et al., 2024), a recent approach, applies evolutionary strategies to search for architectures by leveraging adaptive mutation and crossover operators from Large Language Models (LLMs). We adopt a similar evolutionary search approach as in EvoPrompting, but with a different search space, where our search space is over schemas rather than complex Pytorch operations. As schemas offer a more tractable search space, our method does not require soft prompting and we use GPT-4 out of the box.

## 3 METHODS

ELF consists of Evolution, Language, and Feedback components integrated together for application to a given vision task. In Section 3.1 we overview the vision component of an ELF system, and the MVF modifications necessary to effectively integrate it with the full system. In Section 3.2 we overview the LLM based generation of schemas, which provide the high level contextual biases which are then fed into the vision component. Finally, we overview the evolutionary training component in Section 3.3, whose purpose is to facilitate a process of iterative refinement of schemas, within a communication loop between the language and vision feedback components. The final result of an ELF system is then a schema, refined through evolutionary search to be optimized, and a vision system attuned to that schema such that performance on the vision task is maximized.

### 3.1 MID-VISION FEEDBACK

Mid-Vision Feedback (MVF) is a feedback mechanism designed to bias mid-level feature representations in neural networks towards alignment with high-level context expectations. MVF operates through the application of affine transformations (i.e., linear transformation and possible bias) applied to mid-level features during network runtime. These transformations are context-specific and selectively amplify or dampen certain characteristics.

MVF's training process is divided into two stages. In the first stage, the base network is trained independently, for both the base task and the task of predicting context labels (as per (Maynord et al., 2023)). In the second stage, affine transformations are introduced at the injection level (the level whose features are being modulated), via the feedback loop shown in Figure 2. During back-propagation, gradients are then passed through both the base network and the affine transformations, allowing the entire system to adapt to the context-specific modifications. Training is broken into 2 stages because this produces better performance than training in a single stage - intuitively, this may be because during training, affine transformations operate over a more stable target.

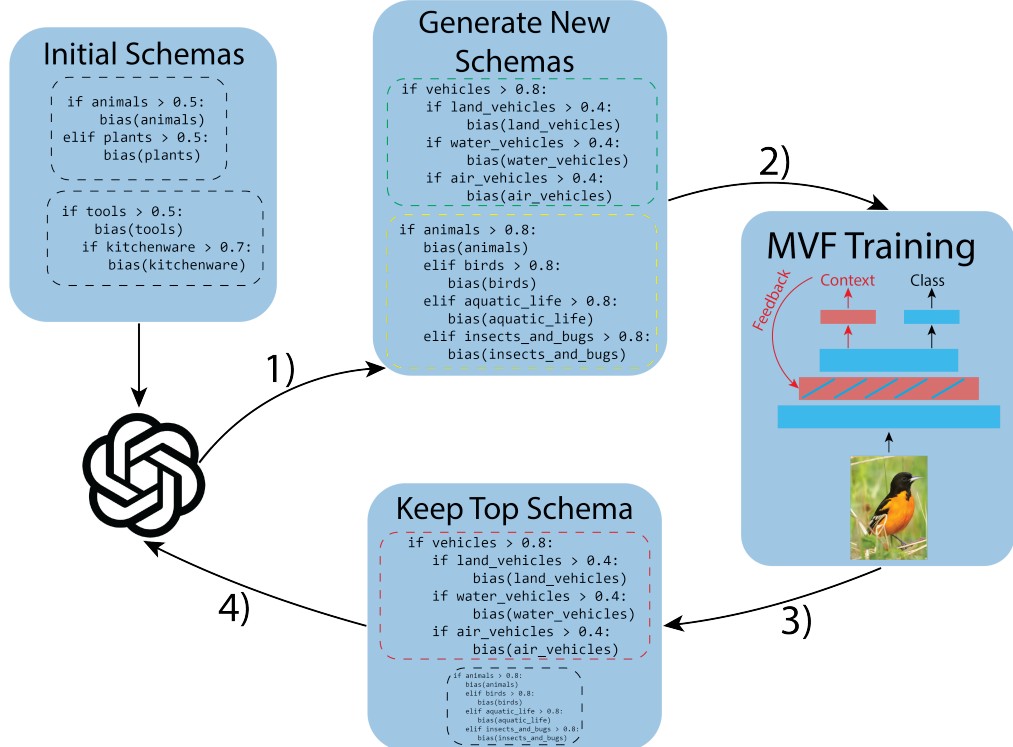

Figure 4: Illustration of the evolutionary search for schemas. After initializing the search with a handful of GPT-4 generated schemas, the meta-learning loop begins. In 1), the LLM synthesizes new schemas from an existing schema set. In 2), the candidate schemas are integrated into MVF and used in both biasing and training the vision network. In 3), we keep the top schemas which produced the highest accuracies. In 4), we send the schemas to the LLM, an the evolutionary process proceeds to the next iteration.

During inference, MVF continues to apply context-specific affine transformations to mid-level features. These transformations are informed by context expectations, which can come from any source, such as those contexts which are predicted by the base network itself and then interpreted through a schema (described in Section 3.2.3). This top-down feedback mechanism ensures that mid-level visual representations are more consistent with high-level context, leading to improved accuracy and contextual consistency.

## 3.2 LANGUAGE

Large Language Models have shown an impressive ability to detect visually useful contexts and attributes for object classification (Menon & Vondrick, 2022), and have proven adept at writing executable code referencing real-world visual entities (Gupta & Kembhavi, 2023). We aim to exploit both of these strengths, using LLMs to generate schemas that function as executable code, and encode contextual information that is useful for object classification.

### 3.2.1 SCHEMA DEFINITION

In our approach, schemas serve as executable structures composed of a (potentially nested) sequence of conditional statements containing context categories. These statements when satisfied invoke context-specific modulations over the mid-level features of the vision network, guided by MVF. See the schema within Figure 5 for an example, where in classifying a bird, contexts such as "nature", "branch", and "sky" might be relevant.

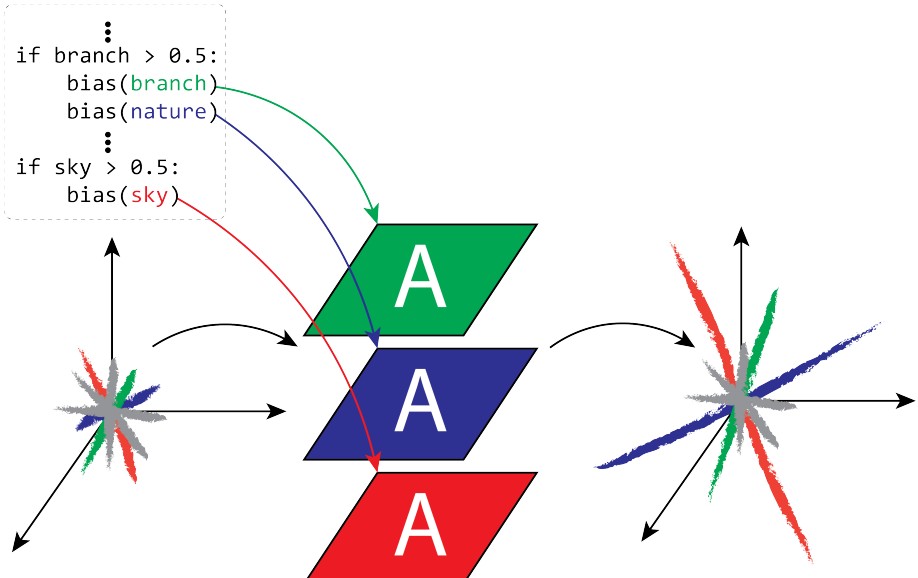

Figure 5: Conceptual illustration of the impact of affine application over injection level features. When it is determined through the schema in use that several contexts should be applied (e.g., branch, nature, sky), then their corresponding affine transformation (i.e., linear transformation and possible bias) are applied over injection level features in the vision architecture (affine transformations are here denoted "A" and color coded according to context). After feature vector modulation the characteristics of the context associated with that affine transformation are more prominent.

### 3.2.2 SCHEMA INITIALIZATION

The initial set of schemas is generated automatically by prompting an LLM with an ontology of object classes from the target vision dataset. These classes, along with a task description (e.g. classify the prominent entity in an image), guide the LLM to produce a set of context categories relevant to differentiating the objects in the dataset. The output of this initialization step is a context ontology, where each object class is mapped to some set of contexts. The initialization also produces some initial schema candidates, creating a foundation for schema refinement. This automatic generation of schemas reduces the need for manual engineering and ensures that the contexts reflect both semantic and visual distinctions relevant to the task at hand.

### 3.2.3 SCHEMA AND MVF INTEGRATION

The integration of our schemas into the vision network requires two forward passes. The initial forward pass of the vision network over an input image produces a distribution over contexts, each with associated probability scores. The names of each context along with the respective probability scores are sent as arguments to the code contained within the schema. This code interprets these arguments to produce a set of contexts with which to modulate the vision architecture through use of MVF. The vision network is then run a second time with this schema modulation to produce object predictions.

See Figure 5 for an example. An image of a Baltimore Oriole is fed in a first pass to the vision network, producing a distribution of context detections. These context detections are fed to a candidate schema, producing the output feedback biases of $\{bias(nature), bias(branch), bias(sky)\}$. We fetch and apply the affine transformations associated with each of these contexts, biasing the features in alignment with the contexts detected.

### 3.3 EVOLUTIONARY TRAINING

Given the candidate schemas provided by GPT-4 at initialization, we use an evolutionary algorithm inspired by EvoPrompting (Chen et al., 2024) to generate improved schemas that best contribute

to the vision task. The primary objective is to iteratively search for optimal schemas that enhance the accuracy. The schema refinement process begins with **A) evaluation** based on classification performance. Promising schemas are then selected for **B) crossover and mutation**. This process is then iterated. This is outlined in Figure 4, and detailed in the rest of this subsection.

**A) Evaluation and Selection:** Each schema is evaluated based on its contribution to object classification accuracy (the "fitness" metric). We integrate each candidate schema into the vision network trained after stage one with MVF, as depicted in Figure 5. After finishing stage two training, the schemas producing the highest accuracies are selected for the next iteration.

**B) Schema Crossover and Mutation:** As in EvoPrompting, the GPT-4 model is employed as an adaptive mutation and crossover operator to produce child schemas. The GPT-4 is responsible for generating these variations, leveraging its pre-trained knowledge of object categories and their relations to produce novel candidate schemas. These crossover and mutation operations allow the exploration of new schemas that might not be obvious through manual design.

Over multiple generations, the schemas are iteratively refined. Each generation is evaluated and tuned, and the schemas with the highest accuracies are preserved for the next generation. This process of mutation and selection iterates until the halting condition is reached: when object classification has not improved for a certain number of generations.

## 4 EXPERIMENTS

In this section we describe our experiments designed to evaluate the effectiveness of ELF. We conducted our evaluations across three datasets — CIFAR100Krizhevsky et al. (2009), ImageNet-1K (Deng et al., 2009), and Caltech101Fei-Fei et al. (2004). We evaluate a total of five models, including scaled-down versions of ResNet20He et al. (2016), ShuffleNet(Sandler et al., 2018), and MobileNet(Zhang et al., 2018), as well as the popular ViT-B/16 (Parmar et al., 2018) and ResNet50 architectures. We present experiments evaluating the extent to which evolutionary methods are suitable for schema discovery (Table 4.1). Experiments are also presented to evaluate the transferability of discovered schemas to different datasets and architectures (Table 4.2). We also assess the distribution of performance benefits across context categories (see Figure 8).

We compute feedback margins - the improvement in performance provided by introducing feedback - as follows: A model consisting only of stage 1 training (no affine transformations or schema feedback) is trained until convergence. A second model with both stage 1 and stage 2 (stage 2 involving affine transformations and schema) is trained until convergence. The feedback margin is then the difference in accuracies between these two models. Note that for both the stage 1 model, and the model trained with stage 2 in addition, context prediction is included in training.

### 4.1 EVOLUTIONARY PROMPTING OVER CIFAR100

We apply the method described in Section 3 over the CIFAR 100 dataset. To make the evolutionary search process tractable, we adopt scaled-down pre-trained vision networks - namely, scaled-down ResNet20, MobileNet, and ShuffleNet - each of which we adapt to take inputs of size $32 \times 32$. Below we list information on the dataset, and implementation details further below.

**CIFAR100** We apply the full ELF search for useful schemas over CIFAR100, a popular dataset of $60,000$ RGB images with $100$ object classes, with $600$ images per class. Our GPT-4 produced initial context ontology consists of $35$ different contexts, some examples of which are *animal*, *vehicle*, *plant*, *cat*, etc. For this initial context ontology, there exists anywhere between $1200$ and $30,000$ images per context. We note that these contexts go beyond the $20$ original superclasses of the CIFAR100 dataset that were hand-engineering.

**Implementation Details** As in the original MVF work (Maynord et al., 2023), we adopt two stages of training, with two separate optimizers. The base network is trained before the beginning of the evolutionary process for $15$ epochs with a base learning rate of $0.001$ and batch size $256$ with an input resolution of size $32$. The evolutionary process takes place starting from weights initialized

after the end of stage one. The search starts with a set of 5 context schema seeds. We run the evolutionary method for 10 rounds with 10 prompts per round, producing 12 candidate schemas per prompt. This means a total of 1200 candidate schemas are generated. However, we apply early stopping, meaning this limit is practically never reached.

|  | Base | Stage 1 | MVF | ELF |
|---|---|---|---|---|
| ResNet20 | 60.66% | 60.43% | 65.12% | **68.13%** |
| ShuffleNet | 61.94% | 60.91% | 63.76% | **68.25%** |
| MobileNet | 64.56% | 64.54% | 65.92% | **70.03%** |

Table 1: Accuracies across a Cifar-100 validation set, for multiple model architectures. We compare the results of ELF against a network involving only stage 1 training (Stage 1) and against a network making use of a single feedback operation as in Maynord et al. (2023) (MVF). We also include results for the model trained without a context classification loss (Base). Stage 1 training is equivalent to base training, with the addition of a context classification loss term. The difference in performance between Base and Stage 1 shows the degree to which accuracy differences are due to the addition of a context loss - the impact is small. (Due to computational limitations the test set numbers are still processing, but the final numbers will be computed by rebuttal period.)

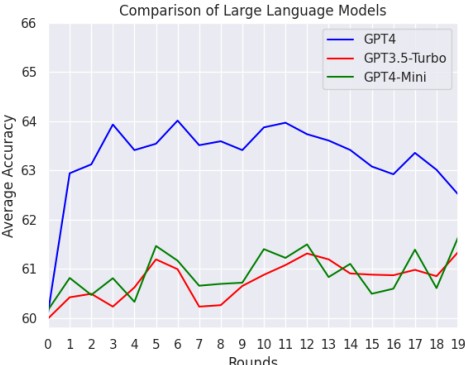

Figure 6: For the purpose of visualization we run the evolutionary search for schemas for 20 rounds (as opposed to the ceiling of 10 rounds performed for Tables 4.1 and 4.2), reporting the average accuracy that an ELF model employing these schemas produces in each round, over multiple GPT models. Numbers are from a ResNet20 model, over the CIFAR100 dataset. Selecting seeds known to perform well or perform poorly had no noticeable impact on the final classification performance after evolutionary search.

## 4.2 TRANSFER TO IMAGENET AND CALTECH101

As ImageNet and Caltech101 are too large to apply evolutionary search over, we take the schema produced by the evolutionary search over CIFAR100, and integrate those schema during the training of models over ImageNet and Caltech101 as in Section 3.2.3. We transfer the schemas produced in Section 4.1 not just to different datasets but to larger models - the ViT-B/16 (Parmar et al., 2018) network and the ResNet50 (Koonce & Koonce, 2021) network. The experimental details are largely identical to those reported in Section 4.1, with the exception of the input resolution - all networks in these experiments ingest images of size 224, and are pre-trained over ImageNet.

**ImageNet** We train over all of ImageNet-1K, and apply GPT-4 to map the ontology of its 1000 classes to the context ontology of 35 contexts which ELF produced over CIFAR100.

**Caltech101** We train over all of Caltech-101, using GPT-4 to map the ontology of its 101 classes to the context ontology of 35 contexts produced by ELF over CIFAR100. The context covering the highest number of classes (*vehicle/machine*) spans 59 classes, and the context spanning the lowest

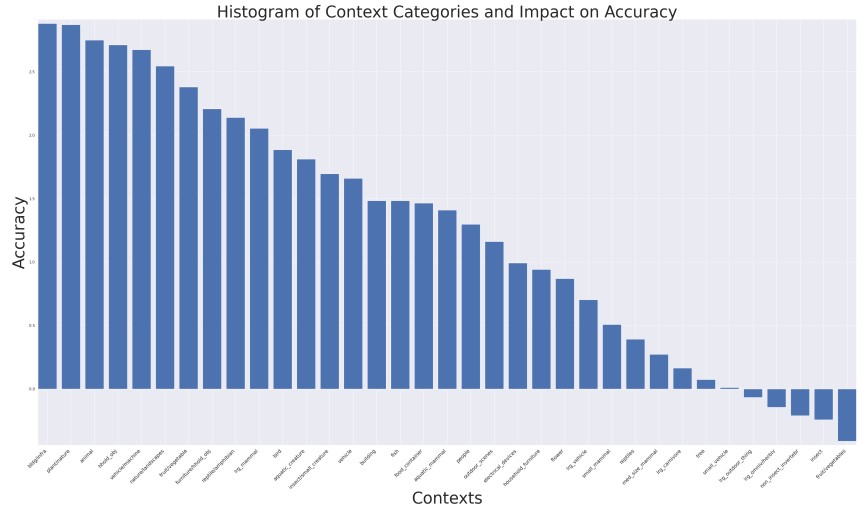

Figure 7: Distribution of accuracy margins (positive implies net benefit to performance, negative implies net damage to performance) where each context is applied on its own as in (Maynord et al., 2023) - that is, each context is applied in isolation (in a separate model) along with its complement (e.g. "animal" and "not animal"). The context is used to bias mid-level features based on its presence in an image, or lack thereof. Numbers are reported for the ResNet20 model over CIFAR100.

| | Caltech101 | | | ImageNet | | |
|---|---|---|---|---|---|---|
| | Stage 1 | MVF | ELF | Stage 1 | MVF | ELF |
| ResNet50 | 83.59% | 84.19% | **85.31%** | 65.90% | 68.5% | **69.01%** |
| ViT-Bx16 | 84.12% | 84.07% | **85.93%** | 70.19% | 70.56% | **72.99%** |
| ViT-Lx16 | 87.47% | 86.72% | **88.71%** | 72.91% | 73.74% | **75.64%** |

Table 2: Accuracies across Caltech101 and ImageNet datasets, for two larger model architectures. In these experiments we transfer the schemas derived using CIFAR100 and the models listed in Table 4.1. We compare the results of ELF against the network after Stage 1 training and against a network making use of a single feedback operation as in Maynord et al. (2023) (MVF). The accuracy increases between Stage 1 and ELF demonstrate the utility that ELF brings to the task of image classification for larger models, and the transferability of schemas to across models and datasets.

number of classes (*air vehicle*) spans 1 class. As the Caltech101 dataset is relatively small in size (anywhere between 40 to 800 images per class), we initialize the network from weights trained over ImageNet (network weights trained over both object and context classification).

## 5 DISCUSSION

We observe in Table 4.1 large performance gains from the introduction of ELF. Our experiments in Table 4.2 demonstrate the robustness and versatility of ELF, showcasing its ability to transfer schemas derived from one model and dataset to different models and datasets while still providing performance improvements.

As seen in Figure 6, with ELF GPT-4 leads to higher average accuracies across the evolutionary search, showing that the schemas generated by GPT-4 are generally more useful than those from older models. We observe GPT-4 exhibits a trend where as the number of rounds increases, the schemas increase in length and complexity, which typically result in decreasing performance.

In Figure 8 we assess the distribution of benefits from the initial GPT-4 generated contexts, derived from CIFAR100, which might be included in a schema. We observe that benefit is not restricted to a small set of contexts, and that the benefits to ELF therefore do not derive from randomly generating

a couple lucky context guesses. From the histogram, a general trend is that the closer to 50% data coverage a context is, the more useful that context is to image classification (e.g. contexts such as *animal* and *nature* are more useful than *insect* and *small vehicle*).

## 6 CONCLUSION

In this paper, we introduce ELF (Evolving LLM-Based Schemas for Mid-Vision Feedback), a framework that combines schema generation using Large Language Models with Mid Vision Feedback for the purpose of improving visual processing. Using an evolutionary method, ELF refines schemas to enhance classification accuracy. Our experiments demonstrate ELF's utility, as well as ELF's adaptability in transfering schemas across architectures and datasets. These findings highlight ELF's potential to bridge language and vision, enhancing visual processing by aligning mid-level visual representations with high-level contextual knowledge.

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

# A APPENDIX

## A.1 ALL CONTEXTS

The contexts adopted in this work are listed as follows:

- aquatic_mammals
- fish_and_marine_life
- flowers
- food_containers
- fruit_and_vegetables
- household_electrical_devices
- household_furniture
- insects
- large_carnivores
- large_man_made_outdoor_things
- large_natural_outdoor_scenes
- large_omnivores_and_herbivores
- medium_sized_mammals
- non_insect_invertebrates
- people
- reptiles
- small_mammals
- trees
- vehicles
- fruit_and_vegetables_1
- insects_and_arachnids
- large_mammals
- reptiles_and_amphibians

- aquatic_creatures
- birds
- vehicles
- buildings_and_man_made_structures
- furniture_and_household_objects
- natural_elements_and_landscapes
- animals
- plants_and_natural_elements
- vehicles_and_machines
- buildings_and_infrastructure
- household_objects
- animate
- inanimate
- nature_and_geology
- tools_and_instruments
- clothing_and_accessories
- food_and_drink

## A.2   QUALITATIVE EXAMPLES

Below we show three example schema programs, along with the corresponding accuracies produced as a result of their incorporation within the ResNet50 architecture over the CIFAR100 dataset.

```
if animate > 0.8:
    bias(animate)
elif inanimate > 0.7:
    bias(inanimate)
```
Accuracy: 61.7%

```
if tools > 0.5:
    bias(tools)
    if kitchenware > 0.7:
        bias(kitchenware)
if birds > 0.5:
    bias(birds)
elif mammals_domestic > 0.5:
    bias(mammals_domestic)
if aquatic_life > 0.5 and mammals_wild > 0.5:
    bias(aquatic_life)
    bias(mammals_wild)
```
Accuracy: 63.9%

```
if animate > 0.5:
    if mammals > 0.6:
        bias(mammals)
    if birds > 0.6:
        bias(birds)
    elif reptiles_and_amphibians > 0.6:
        bias(reptiles_and_amphibians)
    elif insects_and_arachnids > 0.6:
        bias(insects_and_arachnids)
    if fish_and_marine_life > 0.6:
        bias(fish_and_marine_life)
if inanimate > 0.5:
    elif vehicles > 0.6:
        bias(vehicles)
    if furniture_and_household_items > 0.6:
        bias(furniture_and_household_items)
    if nature_and_geology > 0.6:
        bias(nature_and_geology)
    if buildings_and_landmarks > 0.6:
        bias(buildings_and_landmarks)
    if tools_and_instruments > 0.6:
        bias(tools_and_instruments)
    if clothing_and_accessories > 0.6:
        bias(clothing_and_accessories)
    if food_and_drink > 0.6:
        bias(food_and_drink)
```
Accuracy: 66.2%

Figure 8: Three qualitative examples of schemas produced through evolutionary search, with the first (shortest) schema produced earlier in the search, the middle schema produced midway through the search, and the final schema produced near the termination of the search.

## A.3   LIMITATIONS

In the present evaluation we employed datasets with conventional ontologies, with categories familiar to GPT-4. In a highly specialized setting involving specialized manufacturing equipment with which GPT-4 has minimal understanding one challenge would be on seed context generation. One work around to this could be hand engineering seed contexts, or employing an LLM model which

was trained over domain specific text sources. However - as per the next response - we observe that final performance is not highly sensitive to the initial seed context set, so the adaptation to highly specialized domains may require little effort in adjustment.

## A.4 MVF SUMMARY

Mid-Vision Feedback (MVF) is a feedback mechanism designed to enhance mid-level feature representations in convolutional neural networks (CNNs) by aligning them with high-level categorical expectations. This approach incorporates affine transformations and orthogonalization bias to improve contextual consistency and overall accuracy. Affine transformations, defined as

$$\mathbf{f}' = \mathbf{A}\mathbf{f} + \mathbf{b},$$

where $\mathbf{f}$ is the feature vector, $\mathbf{A}$ is the transformation matrix, and $\mathbf{b}$ is the bias vector, are applied to features at designated "injection" levels in the network. These transformations amplify or suppress specific feature characteristics based on the high-level context of the input, which selects the appropriate affine transformation to apply. The context may be derived from an external system or predicted by the network itself through a secondary logits classification head.

The effectiveness of these transformations relies on disentangling features associated with different contexts, achieved through an orthogonalization bias during training. This bias is enforced by a contrastive loss:

$$L_O(F, Y) = \frac{1}{|S_{F,Y}|} \sum_{(f_1, f_2) \in S_{F,Y}} \max\left(0, \frac{f_1 \cdot f_2}{\|\mathbf{f}_1\| \|\mathbf{f}_2\|}\right),$$

where

$$S_{F,Y} = \{(f_1, f_2) \mid f_i \sim U(F_{c_i}), Y_C(f_1) \neq Y_C(f_2), I(f_1) = I(f_2)\},$$

$Y_C(f)$ is the context label for $f$, and $F_{c_i}$ represents features for context $c_i$. This loss penalizes feature similarity across different contexts, encouraging angular separation and making mid-level features more amenable to manipulation through context-driven affine transformations.

Training is divided into two stages. In Stage 1, the base network is trained with the orthogonalization loss to separate features associated with different contexts. The loss function is:

$$L_1(Y, P, F) = \lambda L_O(F, Y) + CE(Y, P),$$

where $CE(Y, P)$ is the cross-entropy classification loss, and $\lambda$ scales the orthogonalization term. In Stage 2, affine transformations are introduced and optimized based on the context of each input. These transformations, initialized as identity matrices with added noise, are trained alongside the network parameters. The loss function for this stage is:

$$L_2(Y, P, F) = CE(Y, P).$$

Affine transformations are selected during runtime by associating each input with a high-level context $c_i$, either predicted by the network or provided as ground truth. These transformations bias mid-level features toward conformity with the selected context, enhancing their relevance for high-level tasks.

MVF exploits CNNs' tendency toward decoupled representations, where feature vector angles correspond to characteristic types and magnitudes reflect variations. By disentangling features across contexts, MVF enables precise control over mid-level representations, bridging the gap between low-level signals and high-level symbolic reasoning. This feedback mechanism outperforms post-hoc filtering by aligning feature representations during runtime rather than discarding inconsistent predictions, achieving superior performance in context-sensitive tasks and allowing for the detection of out-of-context objects.

## A.5 EVOPROMPTING SUMMARY

EvoPrompting is a meta-learning framework that uses large language models (LMs) for code generation as adaptive mutation and crossover operators in an evolutionary neural architecture search (NAS) pipeline. This approach iteratively improves prompts and LM performance through prompt-tuning and evolutionary selection, enabling the discovery of diverse and high-performing neural architectures.

The NAS problem is formalized as searching for architectures $c \in V^*$ (code samples in the LM's vocabulary $V$), evaluated via a fitness function $\text{EVAL}_T(c, D) : V^* \times D \to \mathbb{R}$, where $D$ is a dataset for task $T$. The objective is to maximize the reward function:

$$\arg \max_{C=\{c|c \sim \pi_\theta\}, |C|=k} \mathbb{E}_{c \in C} \left[ \mathbb{E}_{(x,y) \in D}[\text{EVAL}_T(c, D)] \right].$$

To address the intractability of this optimization, EvoPrompting employs a black-box evolutionary algorithm. An LM, $\pi_\theta$, initialized with pre-trained parameters, serves as the mutation and crossover operator, leveraging its pretraining on large code datasets for diverse and semantically valid architectures.

ALGORITHM OVERVIEW

EvoPrompting's end-to-end algorithm consists of the following stages:

1. **Initialization**: The global population $G$ is initialized to an empty list, and the initial population $P$ is seeded with a set of hand-designed architectures $\{c_1, c_2, \ldots, c_p\}$, each evaluated using the fitness function $\text{EVAL}_T(c, D)$.

2. **Crossover and Mutation**: A few-shot prompt is created using $k$ examples randomly selected from $P$. This prompt, combined with task-specific metrics, is used to guide the LM $\pi_\theta$ to generate $n$ child architectures:
$$C = \{c_j \mid c_j \sim \pi_\theta(\cdot|\text{prompt})\}.$$

3. **Evaluation and Filtering**: Child architectures are scored using:
$$\text{fitness}(c) = -\text{model\_size}(c) \times \text{validation\_error}(c),$$
where architectures with validation error exceeding a threshold $\alpha$ are discarded.

4. **Selection and Prompt-Tuning**: The top $p$ architectures with the highest fitness scores are selected to form the new population $P$. The remaining child architectures are used to prompt-tune $\pi_\theta$ for the next round.

5. **Iterative Evolution**: Steps 2–4 are repeated over $T$ rounds of evolution, gradually improving the population and the LM's ability to generate better architectures.

APPLICATIONS AND RESULTS

EvoPrompting was applied to the MNIST-1D dataset and the CLRS Algorithmic Reasoning Benchmark. On MNIST-1D, the algorithm discovered convolutional architectures superior to manually designed baselines, producing smaller models with lower test error. On CLRS, EvoPrompting generated graph neural networks that outperformed state-of-the-art models on 21 out of 30 tasks while maintaining competitive model sizes.

This framework demonstrates the potential of combining LMs with evolutionary techniques to solve complex NAS problems and generalizes beyond architecture design to tasks requiring in-context learning or prompt-tuning.