# OpenReview forum: "Evolved LLM Schemas for Mid Vision Feedback"
_ICLR.cc/2025/Conference — Submitted to ICLR 2025_

### Official Review · Reviewer_hhEf · 2024-10-22

**Soundness:** 3
**Presentation:** 2
**Contribution:** 2
**Rating:** 3
**Confidence:** 3

**Summary:**

The paper proposes a visual learning framework based on mid-vision feedback (MVF) with schema evolution from LLM. The contributions include proposing a new approach to automatically generating the schemas for classes and applying these schemas for image classification with MVF. The experiments also demonstrate that training with the schemas for classification can improve the model performance on CIFAR-100, ImageNet, and Caltech101.

**Strengths:**

1. The idea of using MVF with LLM for generating class schema can be interesting for presenting more accurate ontology for each class instead of obtaining from the rouge output from LLM.

**Weaknesses:**

1. The approach for MVF is hard to follow. Some terms used to define MVF and how to apply it are unclear. For instance, "affine transformation" has often appeared in the paper and should be an important part of applying MVF. However, "affine transformation'" is not clearly defined, making the paper hard to read. Similarly, how the context output from the MVF can be used with the schema should provide more explanation. A recommendation would be adding a small subsection at the beginning of Sec. 3  for introducing how to apply MVF in more detail by defining the terminologies, such as how MVF is trained and also what affine transformation is, and also providing some math formulas for this operation.
2. Some format issues are found. The title for Sec. 2.1 is broken into two lines. Fig. 4 seems to have too many white spaces, and some of the words are not big enough. Making the figure concise by reordering each step and enlarging the word within the block can increase its readability. Some references to the tables do not exist. For instance, Tab. 4.1 is not seen in the paper. Many cited papers have already been accepted by conferences, but the citation is given in the non-updated version.  For instance, the paper from Sachit Menon and Carl Vondrick has been accepted by ICLR23, but the citation has not been updated. Additionally, the conference name is not consistent. For instance, some are written in "In Proceedings of the IEEE conference on computer vision and pattern recognition" and some are written in "In 2016 IEEE Conference on Computer Vision and Pattern Recognition (CVPR)"
3. No explicit schema examples are provided. One of the contributions of this paper is the generation of schemas. However, no demonstration of schemas is found in the paper. Providing some schemas in Sec. 3 or 4 can help readers know more about the ability of this approach. These schemas can be like an image-text pair to show the founding schemas for a specific image.
4. The proposed is simply the combination of MVF and EvoPrompt. No new or dedicated designs are made to combine or apply these two approaches. More clarification for the novelty should be addressed. For instance, discussing why the approach is non-trivial or why the direct combination is optimal.

**Questions:**

1. In Sec. 4, the authors state that the ablation for schemas will be given, but no related sections are found. Is this presented in Sec. 5 and Fig. 6?
2. The models selected as the MVF backbone are not the SOTA ones. Why do the authors adopt these models as the baseline? Can this approach be applied to the current SOTA to enhance performance further?

---

> ### Author Response · Authors · 2024-11-27
> **Reply to Reviewer hhEf**
>
> Thank you for your thoughtful and constructive feedback. We structure our response to you based on the different points you make about improving the work. The points you make in the "Weaknesses" and "Questions" section are fully addressed within the sections below. We have made our paper stronger as a result of your feedback, and have uploaded it.
>
> ***The approach for MVF is hard to follow***
> Thank you for your feedback. We add to the Appendix a more in depth description of the workings of MVF. In it we clarify "affine transformation" in text to make its meaning clear (see lines 53, 206, and 291).
>
> ***Some format issues are found***
> Thank you for catching these! Most of this feedback was used to improve the draft, with the exception of the last. The formatting of the conference name in each case is taken from google scholar, and we use this formatting so that readers may easily use google scholar in finding these papers.
>
>
> ***Example Schemas***
> Thank you for pointing this out. We agree it would be helpful for future readers of the work to see some example schemas to understand the representation. However, schemas are discovered over the entire dataset - not over individual images. As such, we display schemas over the entire CIFAR dataset rather than associated with individual images. Please see Section A3 in the Appendix.
>
> ***Novelty Clarification***
>
>
> To our knowledge we are the first to integrate EvoPrompt and MVF, and more generally, the first to employ high level schema modeling in biasing internal network features in the manner of ELF. This combination is not only technically novel, but conceptually novel: the effect is a bidirectional integration of internal visual representations in CV architectures with semantic contextual modeling from LLMs. As demonstrated through our experiments, the incorporation of EvoPrompt and MVF working jointly improves performance of base networks.
>
>
> ***Ablations***
> Thank you for pointing this out, the ablations referred to were Figures 6 and 7. We make note of this in Section 4 (see 353 - 354).
>
> ***SOTA Models***
> The SOTA models over CIFAR100, Caltech101, and ImageNet are complex ensembles of models, with hyperparameters and training details selected carefully. The usage of feedback within these architectures would warrant extensive experimentation. The models we employ as well as the additional model added in the newer version of the paper make for a total of 6 models over which we demonstrate the utility of ELF. Due to the architecture complexity of the present state-of-the-art models on these datasets, integration into these architectures is beyond the scope of the present work. While experiments over models beyond the 6 already evaluate would strengthen the evaluation even further, we believe that the present experiments present a solid evaluation of the principle of ELF.

---

> > ### Comment · Reviewer_hhEf · 2024-11-28
> > **Official Comment from Reviewer hhEf**
> >
> > I would like to thank the authors for their thoughtful response and clarifications. There are still a few critical issues that remain unresolved. I hope my comments below can help refine the discussion further.
> >
> > > Example Schemas
> >
> > While I understand that schemas can be large and complex, it would be valuable to provide readers with a clear illustration of the top-k schema ordering or selection process. The examples referred to in A2 (not A3, as mentioned) show the selected schema, but they do not clarify how a given schema is determined for a simple image or how the decision process unfolds with the provided schema. Including examples with diverse images and the corresponding schemas, demonstrating the flow of decisions through different if-else statements, would greatly enhance the reader's understanding.
> >
> > > Novelty Clarification and Comparison with SOTA Models
> >
> > I appreciate the authors’ perspective on the novelty of the proposed approach. However, the novelty as presented does not stand out sufficiently in the current paper. While the method is discussed as a tool for schema exploration, the paper focuses primarily on its performance in image classification without elaborating on the properties or implications of the discovered schemas. If the authors aim to highlight performance improvements, it is worth noting that the reported results are still behind the current SOTA models for image classification. Addressing this discrepancy or providing more details on the significance of the schema in contexts beyond performance metrics would strengthen the contribution.

---

> > > ### Author Response · Authors · 2024-12-03
> > >
> > > ***Running Schema Application*** Thank you, we agree this example would provide significant value to the paper. We incorporate a representative example alongside the figure. This example uses the complex schema provided in the same figure, and should give readers an understanding of how the method works.
> > >
> > > ***Schema Novelty*** In evaluating against over-saturated tasks such as image classification over CIFAR and ImageNet, we believe it is not necessary to demonstrate a new-state-of-the-art due to the thousands of papers published each year over these datasets. As for the novelty, we have argued in the Introduction for a biological motivation of schemas, where existing methods fall short, and the gap this work attempts to fill. Hopefully the Introduction argues the significance of the schemas. And the implications of the schemas are that when used, they can be expected to improve performance, as shown in the experiments.

---

### Official Review · Reviewer_GKTC · 2024-10-30

**Soundness:** 2
**Presentation:** 2
**Contribution:** 2
**Rating:** 3
**Confidence:** 4

**Summary:**

In this work, the authors propose ELF (Evolving LLM-Based Schemas for Mid-Vision Feedback), which combines schema evolution with Mid Vision Feedback (MVF). Based on the method of EvoPrompt, the authors utilize the large language model (LLM) as the crossover and mutation operator in the evolutionary algorithm to find a good context split paradigm in Mid-Vision Feedback through the evolutionary algorithm. Finally, the authors conduct experiments on multiple models and data, and give quantitative results, showing the superiority of the overall scheme (ELF) compared with the baseline in the general sense of image classification.

**Strengths:**

1. This paper searches for the optimal schema in Mid Vision Feedback (MVF) through the EvoPrompt method, providing inspiration for the expanded use of EvoPrompt.

**Weaknesses:**

1. The paper combines EvoPrompt and Mid Vision Feedback (MVF), but does not explain the principles and detailed processes of the two in the intrduction or related work section. In addition, the method section is a bit casual, without strict mathematical definitions and rigorous process expressions, making the method not specific and clear enough.
2. The paper does not have sufficient experimental demonstration of the contribution points. There is only an experimental comparison between ELF (the author's method) and the baseline without Mid Vision Feedback (MVF), but no comparison with the image classification result of Mid Vision Feedback (MVF). This does not prove that the schema searched by ELF (the author's method) is better than the schema in Mid Vision Feedback (MVF).
3. The description of the experimental section is not rigorous enough (potentially, it may lead to an imprecise experimental setting). For example, in the comparison of Stage1 and ELF in Table 1, the total training generations of the two do not seem to be consistent. Whether Stage1 has reached sufficient convergence may need to be explained. In lines #346-347, the author mentions using a 32x32 input size neural network for CIFAR100 experiments, but in lines #383-384, the experiment continues on ImageNet, switching to a larger ViT-B/16 and ResNet50, and the resolution setting is not explained at this time.
4. The analysis in the experimental part is not sufficient. The authors can show the difference between the schema optimized by EvoPrompt and the original schema (and MVF), and  explain clearly and more deeply the growth points brought by using EvoPrompt to optimize the schema.

**Questions:**

The writing of the paper needs to be polished, and the format of the paper needs to be improved. For example, the layout of Figure 2 and Figure 3, the pictures and the text should maintain a top-to-bottom relationship. In addition, in Tables 1 and 2, it is not standardized to put the comparison values ​​between different methods in the same column as the methods.

---

> ### Author Response · Authors · 2024-11-27
> **Reply to Reviewer GKTC**
>
> Thank you for your thoughtful and constructive feedback. We structure our response to you based on the different points you make about improving the work. The points you make in the "Weaknesses" and "Questions" section are fully addressed within the sections below. We have made our paper stronger as a result of your feedback, and have uploaded it.
>
> ***EvoPrompt and Mid Vision Feedback (MVF)***
> Thank you for pointing out confusion over the workings of EvoPrompt and MVF - we add to the Appendix summaries of these methods, providing an overview which is more comprehensive than there is space for within the paper. The implementation details of EvoPrompt and MVF are not part of the contribution of ELF.
>
> In the Appendix we make sure to include the equations adopted in the original works (EvoPrompting and MVF) for purposes of clarity and specificity. Please see Appendix sections A4 and A5.
>
> ***Experiments*** Thank you for pointing out the missing baseline with respect to the original MVF work - we extend our experimentation, choosing the single context that produces the highest accuracy using the mechanism in the original Mid-Vision Feedback work - see the updates to Tables 1 and 2.
>
>
> ***Experimental Details*** Thank you for pointing out the missing experimental details. We have included missing experimental details, including that which you pointed out was missing. The input resolution during the evolutionary search is $32\times32$, but when transferring the schemas to new datasets, the ViT and ResNet50 models ingest images of resolution $224\times224$.
>
>
>
>
> ***Schema Optimization***
> Figure 7 displays the benefits of evolutionary search - the seeds produce accuracies hovering around $60\%$, and the accuracies produced by the schemas derived from evolutionary search display sizable increases as the evolutionary search progresses. We hope the new qualitative Figure in the Appendix further provides further illumination (please see A2).
>
> ***Questions:***
>
> Addressing your question, we modify Figure 3, and alter its caption, for clarity. And as per your suggestion, we alter the presentation in Tables 1 and 2, removing the margins columns.

---

### Official Review · Reviewer_5nFs · 2024-11-02

**Soundness:** 4
**Presentation:** 2
**Contribution:** 2
**Rating:** 3
**Confidence:** 4

**Summary:**

This paper presents a method, ELF, to leverage LLMs for mid-vision feedback. The method consists of the automatic production of schemas with GPT-4 and evolutionary search, injecting feedback from these schemas into the middle of a vision network, and producing higher-performing classification networks. The authors report results on a small ResNet, ShuffleNet, MobileNet, and ViT, and analyze performance across LLMs and contexts applied.

**Strengths:**

S1. The method is straightforward and delivers clear boosts in performance. The authors show that MVF with schemas from LLMs has promise for vision tasks, and present evidence across models along with an LLM ablation.

S2. The writing is efficient and concise. Though there is room for improvement in clarity (see below), the authors write with no “fluff” and describe their methods in a straightforward way.

**Weaknesses:**

W1. The writing in this paper is a bit hard to follow, especially the abstract, introduction, and figures.
- The abstract could use some more context in general, especially for some terms (like what a “schema” and “visual learning” is in this context). I can vaguely understand the method being used from reading the abstract (glossing over what an e.g., “semantic program” is), but the actual task being solved is unclear to me. I may just be unfamiliar with this specific subfield of work and common terminology, but in the computer vision space at least, other readers may appreciate a clearer abstract with less unfamiliar jargon.
- The introduction provides some useful information especially in terms of biological motivation, but the problem being solved and the goal task is still unclear. What gap is this method actually addressing?
- Some of the figures are quite confusing to follow, for example:
  - Figure 3. It’s unclear where a reader should start and what path they should follow. Additionally, the function of each arrow is confusing (e.g., what is the output of the evolutionary search and how is it integrated into the other modules it is pointing to?). These details don’t need to be written explicitly in this figure and crowd it if the depiction itself is more straightforward.

W2. The technical implementation of Figure 5 is entirely unclear. How are the contexts extracted from model features, and where do the parameters of the affine transformation come from? How can we be certain that the transformations are being applied to feature projections that cleanly correspond to a single context?

W3. This paper needs more ablations and analyses, including:
- This method seems to require many forward passes per inference, implying an accuracy-efficiency tradeoff. There is no analysis of how the efficiency (commonly measured in FLOPs) scales in training or inference.
- Studying which layer (or combination of layers) responds best when injected with feedback. Is it true that adding feedback to the layers corresponding to “mid-level visual processing” results in the best performance?
- Experiments on larger, more modern models. Consistent boosts across larger CNNs, ConvNeXt models, more ViT sizes, and pre-trained ViTs would provide a more compelling argument for ELF.

**Questions:**

Q1. What is the exact prompt fed into GPT? Pretrained LLMs will know about datasets like CIFAR-10 or ImageNet, so depending on how they are prompted they may recognize the first schema as a collection of CIFAR-10 categories and thus be biased to generate schemas “well-aligned” to the use case. While this can help accuracy in your experiments, the problem is that unfamiliar datasets (e.g., a new satellite image dataset) may not lead to the same boost because GPT will not provide as helpful of schemas.

Q2. Image classification, especially on CIFAR-10 or ImageNet, is a task whose performance is largely saturated by well-trained models these days (like DINOv2, for example). How does this method help for more challenging classification tasks like fine-grained classification, nonstandard domains, or even tasks outside of classification?

Q3. Were these models trained from scratch or fine-tuned?

Apologies if there are some weaknesses in the questions section and vice versa; they tend to blend together when pointing out missing analyses.

Overall, I find this paper to be decently-written and on a good path to be published in the near future. However, there is more work to be done on both the experiment and presentation side to provide a compelling case for ELF in the vision community, and I hope that the authors can take this review as helpful and encouraging feedback to strengthen their paper.

---

> ### Author Response · Authors · 2024-11-27
> **Reply to Reviewer 5nFs**
>
> Thank you for your thoughtful and constructive feedback. We structure our response to you based on the different points you make about improving the work. The points you make in the "Weaknesses" and "Questions" section are fully addressed within the sections below. We have made our paper stronger as a result of your feedback, and have uploaded it.
>
> ***Abstract***
> Thank you for pointing out issues of clarity in the abstract - we update the phrasing to make it clear what these terms are referring to.
>
>
> ***Introduction*** Thank you for pointing out the confusion readers might encounter with the Introduction. We briefly describe the task of image classification approached in this work, and the lack of adoption of context modeling/schemas in image classification. See Lines 63 - 71.
>
>
>
> ***Confusing Figures***
> We rework Figure 3, and the caption of Figure 3, to make the process flow clearer, numbering steps and clarifying associated descriptions. Because Figure 3 involves evolutionary search, which is an iterative rather than single pass process, Figure 3 involves cycles. These cycles may make the beginning and end less apparent as each component is revisited $N$ times in the search.
>
> ***Technical implementation of feedback***
> Thank you for pointing out the lack of clarity. As the details to Figure 5 and the mechanism of feedback require an in-depth understanding of the original MVF paper, we point the reader to the Appendix (A4), where we describe the feedback mechanism in-depth.
>
> Thank you for your questions on Figure 5. These are valid questions, though they are beyond the scope of Figure 5. Figure 5 seeks to illustrate the impact of applying affine transformations over feature vector representations, but to do so without addressing where these affine transformations come from, or how they are trained. This is described in the original MVF paper, and we add a more extensive description of this training process in the Appendix (A4).
>
>
>
>
> ***Flop Analysis***
> There is likely a misunderstanding here, as the forward pass of the vision networks during training/inference requires only two forward passes of the network - the first forward pass to predict context, and the second forward pass to do classification. The details of this are contained within the Appendix.
>
> While the question of computational complexity in the training process is an interesting one, computationally efficient training is not the objective of ELF, nor in general the objective of evolutionary search algorithms. ELF seeks to maximize test time performance, while not adding computational burden during test time.
>
> ***Injection Level***
> The optimal injection level for the feedback mechanism was taken directly from the original feedback paper, where the penultimate layer was chosen across all architectures. As the feedback mechanism used in ELF is identical to that of MVF, we adopt this injection level and refer the reader to MVF for the accuracies over different injection levels.
>
> ***Experiments on larger, more modern models***
> Present experiments across 5 different architectures demonstrate consistent benefit to ELF. These include a range of architectures, including both CNN and transformer architectures. We include one more set of results adopting a state-of-the-art, ViT-L network pre-trained over ImageNet. Please see Table 2. We note however that both ResNet50 and ViT-Bx16 networks are already pre-trained, and make mentions to this in the Experiments section.
>
> ***Exact Prompt***
> Thank you for this question - we include the prompt in exactness in the Appendix (please see A6). In the prompt, we do not name the dataset, but rather only the list classes constituting the dataset. This prevents the network from dataset-specific bias.
>
> We also observe that final ELF performance is not highly sensitive to initial context seed set (see response to Reviewer 1), and as such a priori familiarity with the dataset in question would not be required of GPT-4. Also consider the generic nature of produced contexts: context seeds sets are discernable from the object list of a dataset, and are not specific to the dataset.
>
> ***Other Vision Tasks***
> Feedback has been shown to help in the task of video classification, and we believe can benefit networks trained over the task of image segmentation. As such, there is a strong case for the application of ELF in both of these tasks. We leave this to future work.
>
> ***Pre-training***
> All networks are pre-trained over ImageNet. We make a note of this in the Experiments section.

---

### Official Review · Reviewer_d6Kj · 2024-11-04

**Soundness:** 3
**Presentation:** 3
**Contribution:** 3
**Rating:** 6
**Confidence:** 4

**Summary:**

- The paper presents ELF (Evolving LLM-Based Schemas for Mid-Vision Feedback), a framework integrating schema evolution with Mid-Vision Feedback (MVF) for enhanced visual learning.
- ELF uses Large Language Models (LLMs) to automatically generate executable schemas—semantic programs based on contextual categories that inform visual processing.
- The framework employs EvoPrompt, an evolutionary algorithm, to iteratively refine schemas for improved classification accuracy and contextual alignment.
- ELF demonstrates effectiveness across multiple datasets and models, showing that evolved schemas can be transferred to different architectures, maintaining performance improvements.
- The approach emphasizes biologically inspired feedback, where high-level context guides mid-level feature representation, mirroring how biological vision systems operate.
- The work highlights ELF's adaptability and potential for bridging vision and language, enhancing visual processing by aligning representations with task-specific contextual knowledge.

**Strengths:**

- The paper presents originality through its development of ELF, a framework that uniquely combines LLMs with Mid-Vision Feedback (MVF) to mimic biological vision systems, enhancing visual learning by integrating high-level contextual knowledge into mid-level feature processing.
- The use of EvoPrompt for schema generation and iterative refinement demonstrates a creative application of evolutionary algorithms, expanding the role of LLMs beyond traditional text-based tasks into schema-based visual learning.
- The quality of the research is supported by comprehensive experiments across CIFAR100, ImageNet, and Caltech101, showing consistent improvements in accuracy and schema transferability across different architectures. The methodology is detailed enough to ensure reproducibility.
- Clarity is maintained throughout the paper, with well-organized sections and the effective use of diagrams to illustrate complex processes such as schema generation and MVF integration. This helps readers grasp the approach and its underlying mechanisms.
- The significance of the work lies in its contribution to the field of vision-language integration, demonstrating how LLM-generated schemas can be used to guide and improve visual processing. The framework shows potential for wider application in various vision tasks that require contextual adjustments.
- ELF’s demonstrated ability to transfer schemas across datasets and architectures underscores its adaptability and robustness, making it a promising tool for advancing vision systems in real-world scenarios and opening doors for further exploration in schema-driven learning.

**Weaknesses:**

- The paper's focus on a limited set of visual contexts (e.g., animate/inanimate, nature/urban) restricts the generalizability of findings. Expanding experiments to include a wider variety of context types would enhance the applicability and depth of the results.
- Transferability across more diverse datasets is not fully explored. While the paper shows results for CIFAR100, ImageNet, and Caltech101, adding experiments on more challenging datasets with higher intra-class variability could strengthen claims of generalizability.
- The paper lacks a thorough comparison with other context-modulating techniques beyond its scope (e.g., attention-based feedback mechanisms). Including a comparison with alternative top-down or context-driven approaches would offer clearer insights into ELF’s relative strengths and weaknesses.
- Analysis of failure cases is minimal. Presenting specific examples where ELF did not perform well or where the schema approach struggled would provide valuable insights for future iterations and improvements.

**Questions:**

- Could you elaborate on how the complexity of the generated schemas affects interpretability and practical use? Are simpler schemas equally effective or is there a trade-off between complexity and performance?
-  Do you anticipate any challenges in integrating ELF with existing vision systems in industry settings? What adjustments or considerations would be necessary to adapt ELF for these scenarios?
- How critical is the initial set of schemas generated by GPT-4 on the overall results? Would the use of a different LLM or a smaller language model significantly impact ELF's effectiveness?

---

> ### Author Response · Authors · 2024-11-27
> **Reply to Reviewer d6Kj**
>
> Thank you for your thoughtful and constructive feedback. We structure our response to you based on the different points you make about improving the work. The points you make in the "Weaknesses" and "Questions" section are fully addressed within the sections below. We have made our paper stronger as a result of your feedback, and have uploaded it.
>
> ***Context limitations*** - Rather than the context ontology used in this work being limited, our ontology includes $26$ different contexts. We include the full list in the updated Appendix section, in Section A1.
>
> ***Transferability*** - The transferability of schemas across datasets is one advantage of ELF. As schemas are domain knowledge representations, they are transferable to the extent that domains are shared across datasets. Many of the contexts selected by ELF are general in nature (e.g., animate/inanimate, nature/urban, as you point out), and as such are generalizable across a wide range of datasets. However, contexts produced for one dataset will not generalize to a different dataset with no domain overlap - for example, schemas produced for a dataset of birds will not generalize to a dataset of manufacturing settings.
>
> The present ELF contexts have high intraclass variability (e.g., all inanimate objects is a class with high intra-class variability), which allows it to generalize across datasets. Datasets which share no structure cannot be generalized across. Note that the object ontologies of the datasets evaluated differ.
>
> A more extensive consideration across more than the 3 datasets evaluated could provide some benefit, though we believe that the demonstrated generalizability of ELF models across these datasets effectively demonstrates that the principle of ELF is solid.
>
>
>
>
> ***Comparison to other context modulation works*** - Thank you for pointing out this missing baseline. We incorporate the context modulation approach from MVF, choosing the single context in the ontology that produces the highest accuracy. Please refer to the baseline MVF in Tables 1 and 2.
>
> ***Failure cases*** - Thank you for pointing this out. We agree it would be helpful for future readers of the work to see some example schemas to understand the representation. Please see the newly added figure in the Supplementary Materials (A2), where we also address your first question. Typically, more complex schemas perform better within ELF while being less interpretable than the simpler schemas.
>
> ***Industry Applicability*** - In the present evaluation we employed datasets with conventional ontologies, with categories familiar to GPT-4. In a highly specialized setting involving specialized manufacturing equipment with which GPT-4 has minimal understanding one challenge would be on seed context generation. One work around to this could be hand engineering seed contexts, or employing an LLM model which was trained over domain specific text sources. However - as per the next response - we observe that final performance is not highly sensitive to the initial seed context set, so the adaptation to highly specialized domains may require little effort in adjustment. We add this content to a separate "Limitations" subsection within the Appendix (A3).
>
> ***Different Seeds, different LLMs*** - We found the question of seed selection interesting to explore as well. We observed no noticeable impact on the final performance whether the seeds were handwritten or the seeds were automatically generated by GPT4. Furthermore, selecting seeds known to perform well or perform poorly had no noticeable impact on the final classification performance after evolutionary search. We add a short note to the caption of Figure 6 mentioning this. As for the question of whether different LLMs perform better, please see Figure 6 where we compare GPT4, GPT3.5-Turbo, and GPT4-Mini. The choice of LLM makes a pronounced difference in final classification results.

---

### Meta-Review · Area_Chair_7CvN · 2024-12-16

**Metareview:**

The paper introduces a framework designed to enhance visual learning by integrating schema evolution with Mid Vision Feedback (MVF). The authors propose to use LLMs to automatically generate executable semantic programs (schemas) which operate over different context categories. These schemas allow top-down feedback connections that enrich mid-level visual processing with high-level contextual information. The authors employ EvoPrompt, an evolutionary algorithm that refines schemas via iterative search, leading to increased accuracy and contextual consistency.

The reviewer consensus on this submission is clear, with reviewers citing:
+ Focus on a narrow set of visual contexts, which significantly limits the generalizability of findings, as well as a lack of thorough comparisons with existing context-modulating techniques and minimal analysis of failure cases.

+ Insufficient writing, as seen in the paper abstract in the original submission, the poorly presented figures that hinder comprehension, and the lack of clear definitions of key technical concepts such as context extraction methods and the parameters of affine transformations. Moreover, issues with formatting, citation inconsistency, and a lack of explicit schema examples further inhibit reader comprehension and detract from the paper's overall quality.

+ Insufficient ablation studies and analyses, particularly regarding efficiency, the effect of feedback on different layers, and the scalability of results to larger models.

Although this paper proposes some interesting ideas, the novel contributions of the proposed framework remain unclear. In rebuttal, the authors clarified some critical points and addressed some of the formatting issues, however the manuscript is in need of significant revision in form and content before it can be considered for publication.

**Additional Comments On Reviewer Discussion:**

The reviewer/author discussion period consisted mostly of the authors providing clarifications on key points of confusion on the part of reviewers. Presentation quality and technical clarity, mentioned by the majority of reviewers as significant, were the main points leading to the Reject decision.

---

### Decision · Program_Chairs · 2025-01-22

Reject